# Disinhibition enables vocal repertoire expansion after a critical period

Fabian Heim [1], Ezequiel Mendoza [1,3], Avani Koparkar [1,2,4] & Daniela Vallentin [1] ✉

The efficiency of motor skill acquisition is age-dependent, making it increasingly challenging to learn complex manoeuvres later in life. Zebra finches, for instance, acquire a complex vocal motor programme during a developmental critical period after which the learned song is essentially impervious to modification. Although inhibitory interneurons are implicated in critical period closure, it is unclear whether manipulating them can reopen heightened motor plasticity windows. Using pharmacology and a cell-type specific optogenetic approach, we manipulated inhibitory neuron activity in a premotor area of adult zebra finches beyond their critical period. When exposed to auditory stimulation in the form of novel songs, manipulated birds added new vocal syllables to their stable song sequence. By lifting inhibition in a premotor area during sensory experience, we reintroduced vocal plasticity, promoting an expansion of the syllable repertoire without compromising pre-existing song production. Our findings provide insights into motor skill learning capacities, offer potential for motor recovery after injury, and suggest avenues for treating neurodevelopmental disorders involving inhibitory dysfunctions.

We acquire speech by listening to and imitating vocalizations heard in our environment during development[1–3]. Although speech sounds from novel languages can be learned later in life, the ability to accurately reproduce non-native speech sounds is age-dependent[4,5]. The decline of vocal learning capabilities has been associated with a decrease in brain plasticity leading to reduced vocal flexibility[6]. In many animals, vocal imitation learning is restricted to a critical period after which the vocal repertoire remains largely unchanged[7–9]. The closure of this vocal-motor critical period has been linked to a decrease in synaptic plasticity[10], the stabilization of perineuronal nets[11,12], structured changes to the inhibitory circuitry[13,14] and a change in intrinsic properties of neurons in vocal learning-relevant brain areas[15]. Critical periods also exist for sensory modalities[16] associated with auditory perception[17,18] and visual processing[19] underlying behavioural imprinting[20]. In a few cases, it has been demonstrated that sensory critical periods can be reopened by targeted manipulations of the corresponding circuitry[21,22]. Here, we asked whether the inhibitory

factors that contribute to the closure of a vocal motor critical period can be circumvented, leading to the recovery of behavioural plasticity in older animals.

Zebra finches (*Taeniopygia guttata*) learn their songs from adult males within a critical period of 90 days post hatch[23] after which the song remains mostly unchanged[24,25]. The song system of male zebra finches consists of a network of discrete brain regions called nuclei[26]. The nucleus HVC (proper name) receives auditory input important for vocal imitation learning[27–31] and plays a role in vocal motor control[32–34]. GABAergic inhibitory interneurons in HVC of zebra finches modulate their activity during song learning, exhibiting heightened activity when the bird listens to familiar tutor song syllables, but not novel ones[13]. This suggests that inhibition within HVC plays an important role in ending the critical period for vocal learning, potentially by controlling auditory-induced changes in excitatory premotor neurons (see Fig. S10 in ref. 13). We predict that suppressing the influence of these interneurons in adult zebra

[1]Max Planck Institute for Biological Intelligence, Seewiesen, Germany. [2]Indian Institute of Science Education and Research (IISER), Pune, India. [3]Present address: Freie Universität Berlin, Berlin, Germany. [4]Present address: Eberhard-Karls-Universität Tübingen, Tübingen, Germany. ✉ e-mail: daniela.vallentin@bi.mpg.de

finches could re-open the critical period, allowing them to learn new syllables.

By manipulating the activity of HVC inhibitory interneurons bilaterally, we aim to overcome one constraint imposed by the end of the critical period and enable adult zebra finches to produce new song syllables.

## Results

### Pharmacologically lifted inhibition in HVC alters the adult song

The song of an adult male zebra finch becomes more stereotyped over the course of its life[25] unless the animal experiences sensory disturbances such as deafening[35,36]. However, it has been shown that birds can alter the pitch of syllables or the syllable sequence when exposed to distorted auditory feedback over extended periods[37,38]. To test whether the passive exposure to auditory stimuli over a short time scale (for 1 h) is sufficient to induce changes in song production we presented adult birds ($n = 5$ birds) with playback of the bird's own song (BOS) which had been modified with respect to their syllable sequence (Fig. 1a). Concurrent bilateral infusion of phosphate-buffered saline (PBS) did not affect singing[39]. Birds did not change the sequence of their song towards the sequence presented during playback, nor were spectral features altered (Fig. 1b and Supplementary Fig. 1a). To reduce the impact of inhibition we then combined the same playback with the application of gabazine, a GABA-A receptor antagonist, on HVC to test whether inhibition is necessary to preserve a stable song in the presence of external auditory input. We found that birds produced modified vocalisations (Fig. 1b, c and Supplementary Fig. 1b) when the impact of inhibition within HVC was limited. To quantify changes in vocal output we calculated the self-similarity across renditions for each pre-existing syllable produced in the control condition and the similarity of all syllables and the modified syllables produced in the gabazine condition. We considered a syllable as novel when it was significantly different from all other syllables (Fig. 1d, see 'Methods') and found that under gabazine conditions all birds added one or two novel syllables (Fig. 1e–g). Additionally, the self-similarity of individual syllables was reduced when inhibition was lifted (Fig. 1c, f)[39]. To understand when the novel syllables were produced within a song motif, we calculated transition probabilities of all syllables under control and gabazine conditions. We found that song stereotypy was stable during control conditions whereas under gabazine novel syllables were sung at variable positions (Fig. 1h). Next, we inquired how the addition of novel syllables changed the overall song structure and analysed the persistence of the syllable progression (song linearity), and the reoccurrence of a specific syllable sequence across renditions (song consistency)[40]. Both linearity (Fig. 1i) and consistency (Fig. 1j) of the song were reduced but not significantly so, when gabazine was applied (Supplementary Fig. 1f, g). To determine if the song changes were solely dependent upon the presence of gabazine in HVC while the birds produced song or whether the auditory input also contributed to the alterations, we applied gabazine without playback stimulation (Supplementary Fig. 1c). Two out of four birds (Supplementary Fig. 1d) that received gabazine infusions without playback produced novel syllables. However, similar to the song of birds that received playback, the song parameters measured as the differences in spectral features to the baseline song (Supplementary Fig. 1a, b) or linearity and consistency of song production were not significantly different from control birds (Supplementary Fig. 1e–h).

These findings indicate that disinhibition within the HVC of adult zebra finches enables the production of modified syllables, which can appear within the context of an established syllable sequence[39]. However, temporally confined disinhibition may not cause lasting changes in the HVC circuitry. Instead, it likely uncovers hidden excitatory inputs, leading to additional vocalisations that would normally be suppressed by inhibition. This suggests that normally occurring inhibition within HVC is important for actively constraining the adult vocal repertoire and the motif-level structure of the song.

### Targeted optogenetic manipulation of inhibitory interneurons in HVC

Although the application of gabazine is specific in terms of its binding affinity for and inactivation of GABA$_A$ receptors[34,41] (Supplementary Fig. 2) its effect cannot be precisely controlled on the time scale of individual zebra finch song. In the aforementioned paradigm, gabazine lifts inhibition during both playback exposure and song production. Therefore, in our interpretation of the observed song alterations, the relative contributions of auditory vs vocal motor disinhibition cannot be clearly disentangled. To circumvent this potential confound, we aimed to restrict our perturbation to HVC interneuron activity exclusively during the presentation of song stimuli.

To target GABAergic inhibitory interneurons specifically, we designed an AAV-based construct including the enhancer mDlx which has been shown to target GABAergic neurons within HVC[42] expressing either only a fluorophore (AAV2/9-mDlx-eYFP, control, Fig. 2A) or an additional hyperpolarising channel archaerhodopsin ArchT (AAV2/9-m mDlx-ArchT-eYFP, ArchT+, Fig. 2B).

First, we confirmed the virus' labelling efficiency, specificity, and its functionality. The labelling efficiency of the viral construct as measured by corrected total cell fluorescence (CTCF)[43] did not differ between the HVC of birds injected with the control virus (Fig. 2A, CTCF = 2611 ± 317) and the HVC of birds injected with the ArchT-expressing construct (Fig. 2B, CTCF = 2148 ± 290, $N = 6$ birds). The specificity, i.e. the median overlap of interneurons expressing parvalbumin (PV) and eYFP in HVC was similar between control and ArchT+ birds (Fig. 2C). In addition to PV, subtypes of interneurons in HVC also express calbindin (CB) and calretinin (CR)[44–46]. When testing the overlap of virally transduced cells that express PV, CB, or CR we found that the overlap of ArchT/eYFP increased to 81.2 ± 7.8% ($n = 1$ bird) which is similar to previously reported results based on a virus with the same promotor expressing only GFP[42].

To validate the constructs' functionality, we performed extracellular recordings in the HVC of control and ArchT+ birds. We confirmed that the recording sites were located in the vicinity of transduced interneurons by post hoc reconstructions of the probe tract (Fig. 2D). Single units were sorted according to their waveforms and spiking activity was aligned to the onset of light stimuli. In control birds, we did not find neurons that changed their activity profile in relation to light stimulation (Fig. 2E, 0/198 units, $n = 3$ birds, 5 hemispheres). In ArchT+ birds, we found 6.41% of the recorded neurons were transiently silenced (Fig. 2F, 5/78 units, $n = 2$ birds, 4 hemispheres) and 15.39% were transiently disinhibited during optogenetic stimulations (15/78 units, Supplementary Fig. 3). Although previous studies suggest that about 10% of HVC neurons are interneurons[44,46] the underrepresentation of silenced interneurons during our recordings might be explained by the viral approach lacking complete transduction of all interneurons and the possibility of stimulation light attenuation in the more ventral portions of HVC. Despite these limitations, this method enabled us to selectively decrease the activity of HVC interneurons in a temporally precise manner to test if the disinhibition of HVC during auditory experience leads to a reintroduction of vocal plasticity in adult zebra finches.

### Optogenetic silencing of HVC interneurons during playbacks leads to the production of novel song syllables in adult zebra finches

Once the viral construct was validated in vivo, we tested whether the precise inactivation of HVC interneuron activity during exposure to a novel song would re-establish juvenile-like disinhibited conditions on this premotor circuit and enable auditory-driven changes in adult zebra finch songs. Therefore, birds underwent bilateral virus injections

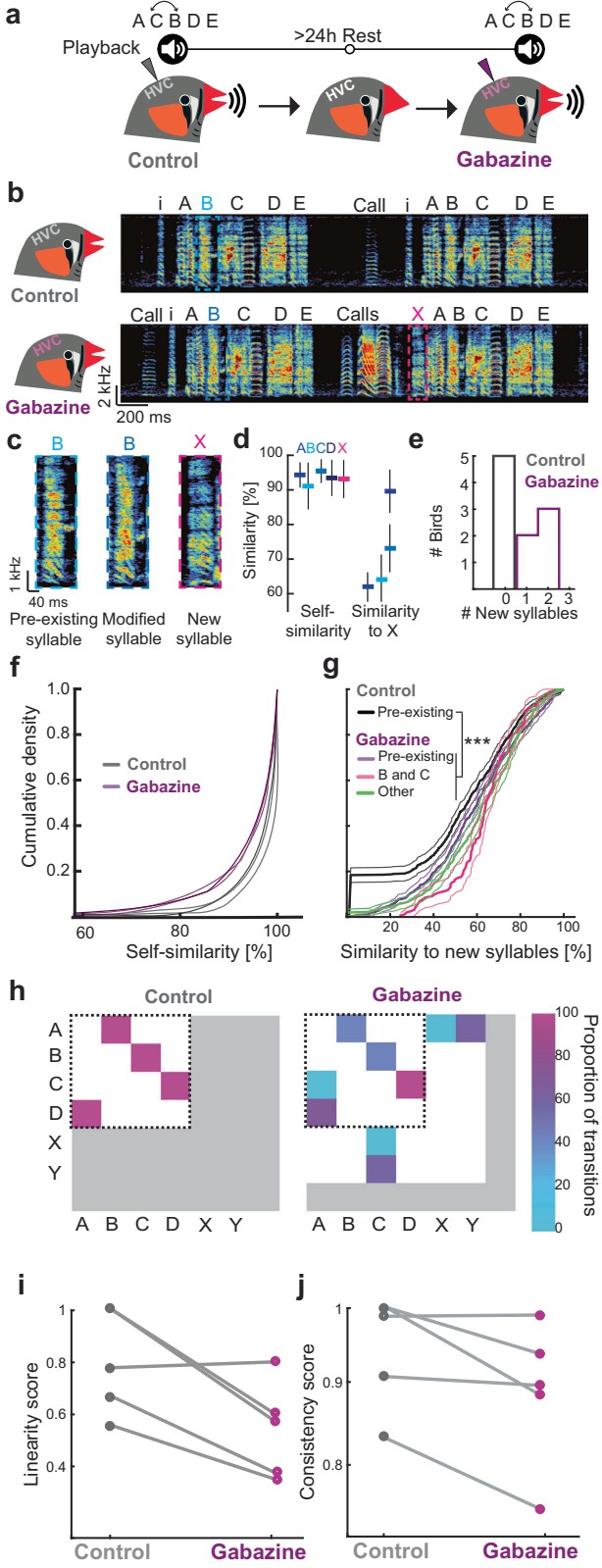

**Fig. 1 | Pharmacological limiting the impact of inhibition in HVC of adult zebra finches leads to the production of altered songs. a** Adult zebra finch song was recorded while individual birds were exposed to playbacks of their own song with a switched order of syllables B and C and PBS was bilaterally infused in HVC. After a 24-h rest period, the experiment was repeated while gabazine was applied. **b** Example song bout of a bird during PBS- (top), and during gabazine-infusion (bottom). 'i' indicates introductory notes. Capitalized letters 'A–X' indicate different syllables. **c** Examples of a syllable 'B' produced during control, the same syllable under gabazine condition and a new syllable that emerged during gabazine application, highlighted in (**b**). **d** Self-similarity scores (mean ± std) across ten renditions of pre-existing syllables and similarity score for each pre-existing syllable with respect to a novel syllable ('X') produced by a bird during gabazine infusion. **e** Number of new syllables produced during control condition (PBS infusion (grey)) and during gabazine infusion (purple). **f** Empirical cumulative density (ECD) curves of self-similarity comparisons of all syllables produced by control and gabazine-infused birds. Bold lines highlight the means, thin lines indicate the 95% confidence interval for each data set. **g** ECD curves and 95% confidence intervals of cross-similarity comparisons between pre-existing and novel syllables for control (black/grey) and gabazine (dark/light purple) conditions. Pink/light pink lines represent comparisons between syllables B and C and the new syllable. Green/light green lines represent comparisons between all other syllables and the new syllable. Bold lines highlight the means, thin lines indicate the 95% confidence interval for each data set. **h** Matrix plots illustrating syllable transition proportions. An exemplary bird sings a stereotyped syllable sequence ABCDABCD (left). This core motif sequence remains intact after gabazine infusion (right). Novel transitions leading to new syllables are added towards the beginning or in the core motif's centre as indicated by the horizontal and vertical arrangement of rare and new transitions following the core syllables outside of the range covered before gabazine infusion (dotted black line). **i** Song linearity during the control or gabazine condition for ten motifs from N = 5 birds each. **j** Song consistency score during the control or gabazine condition for ten motifs from N = 5 birds each. Source data for (**d**, **i**, and **j**) are provided as a Source Data file.

weeks after the stimulation phase had ended, the final song of the birds was recorded and the birds were then sacrificed to validate virus expression and fibre placement.

During final song recordings, ArchT+ birds produced novel syllables that were absent during their baseline songs (Fig. 3b, c and Supplementary Figs. 4a, b and 5). According to feature-based similarity measures[48] the novel syllables differed acoustically from previously produced syllables and syllables of the song playback (Fig. 3a–d). All ArchT+ birds produced new syllables whereas none of the control birds exhibited expanded repertoires (Fig. 3e). Aside from the addition of novel syllables, core motif sequences and spectral properties of pre-existing syllables remained unchanged as determined by similarity analyses (Supplementary Figs. 4d and 5b, c).

To identify novel syllables with an unbiased approach we employed a supervised deep neural network (deep audio segmenter (DAS)[49]) and trained it with all syllables from ten motifs that were recorded from each control and ArchT+ birds during baseline to obtain a model that recognized zebra finch syllables. We then used the trained model to segment all syllables produced during baseline and final song recordings. To identify different syllables based on the annotations predicted by DAS, we employed uniform manifold approximation and projection (UMAP) dimensionality reduction and identified syllable clusters by employing the density clustering algorithm HDBscan[50,51] (Fig. 3f, g and Supplementary Fig. 6). Control birds' syllables and syllables recorded during the final song recording mostly clustered together indicating that the originally produced syllables (pre-existing syllables) remained unaffected by light stimulation combined with novel song playback. However, for each ArchT+ bird (but none of the control birds) new syllable clusters emerged after optogenetic stimulation and the numbers of clusters closely corresponded to the numbers of new syllables with the feature-based method (feature-based = 2–7 new syllables, HDBscan clustering = 2–5 new clusters) indicating that acoustically

into HVC of either the ArchT+ (N = 6 birds) or the control construct (N = 4 birds) and were implanted with optic fibres above HVC bilaterally. Six weeks after injections directed song was recorded to serve as a baseline (Fig. 3a). The following stimulation phase lasted four weeks and consisted of 200 daily playbacks of an artificially created zebra finch song[47] (Fig. 3a). Each playback iteration was coupled with optogenetically induced silencing of ArchT-transduced interneurons. Four

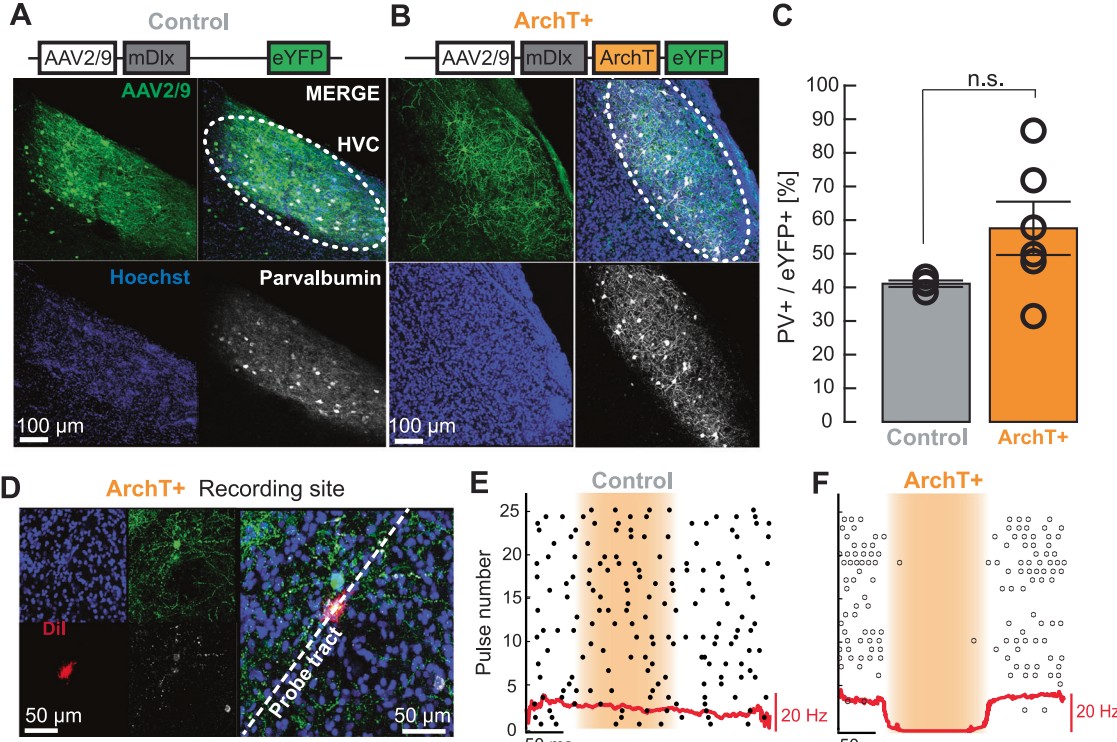

**Fig. 2 | Viral strategy to optogenetically silence inhibitory interneurons in zebra finch HVC. A** Adult zebra finches were transfected with an AAV-based construct injected into HVC, bilaterally. Representative example of eYFP expression (green) and co-localization (merge) with the inhibitory interneuron marker parvalbumin (white) in HVC. Cell nuclei were stained with Hoechst (blue). $N = 4$ birds, $p = 0.413$, Wilcoxon rank sum test, Scalebar: 100 μm. **B** Representative example of ArchT-eYFP expression in HVC. **C** Quantification of co-localized AAV-mediated eYFP and PV. Overlap (control) = 41.3%, $n = 4$ birds, overlap (ArchT+) = 53.8%, $N = 6$ birds; $p = 0.1142$, Wilcoxon rank sum test. Error bars indicate the standard error of the mean. Scalebar: 100 μm. Source data are provided as a Source Data file. **D** Recording site in the vicinity of an ArchT-transduced inhibitory interneuron expressing both eYFP (green) and PV (white). The multi-electrode probe tract was coated with fluorescing DiI (red) for tract reconstruction (dotted line; blue counterstain = Hoechst). Scalebar: 50 μm. **E** Spike raster plot for an exemplary unit in HVC aligned to pulses of light stimulation (yellow shaded area), in a bird injected with the control construct. The red line is the peri stimulus time histogram. **F** Spike raster plot of a unit in HVC of an ArchT-expressing bird, exhibiting suppression of firing activity during pulses of optogenetic stimulation (yellow shaded area).

coupled disinhibition of HVC enables novel syllable production in adult birds.

To rule out the possibility that the new song syllables are just the co-occurrence of other vocalizations which zebra finches produce during bonding and social interaction outside of the context of singing[52–54] we compared the birds' calls and their introductory notes with the newly identified syllables. New syllables were not similar to calls or introductory notes produced by the same birds (Supplementary Figs. 5 and 6A, $78.13 \pm 20.8\%$ similarity; tet, stack, long calls, introductory notes). These findings indicate that novel syllables are not derived from the birds' call repertoire.

Next, we tested whether the new syllables produced by ArchT+ birds were stable across renditions as is typical for zebra finch song[23]. By assessing the self-similarity of pre-existing and new syllables of control and ArchT+ birds we found that pre-existing syllables remained stereotyped after treatment as indicated by high self-similarity scores (Fig. 3h, blue curve, self-similarity(control) = $93.04 \pm 14.56\%$, grey curve, self-similarity(ArchT+) = $94.75 \pm 6.34\%$) whereas new syllables were less similar across repetitions (yellow curve, self-similarity(ArchT+ new) = $85.59 \pm 15.06\%$) (Fig. 3h). Self-similarity was significantly different between the different syllables. This implies that although average similarities of novel syllables fall within the range of pre-existing syllables of ArchT+ birds (Fig. 3h), novel syllables are more variable than pre-existing syllables.

To investigate whether the new syllables are similar or different from the pre-existing syllables and from the playback, we performed cross-similarity comparisons (Fig. 3i). We found that, the cross-similarity scores between pre-existing and new syllables (Fig. 3i, yellow curve, $52.77 \pm 24.55\%$) were significantly higher than comparisons between new syllables of ArchT+ birds and those of the playback stimuli (Fig. 3i, purple curve, $36.35 \pm 12.43\%$). Furthermore, cross-similarities for all syllables were lower than the self-similarity of these syllables.

Taken together, similarity comparisons showed that the variability of novel syllables produced by ArchT+ birds was higher compared to pre-existing syllables sung by both, ArchT+ and control birds. Additionally, novel syllables differed from all other zebra finch vocalizations that were produced by the tested birds. However, new syllables produced by ArchT+ birds were generally more similar to pre-existing syllables than playback syllables. This result might be interpreted in different ways. It might be suggestive of the existence of physical song production constraints that were acquired during the critical period for song learning or, alternatively, it might be a result of a discontinued song imitation process at a certain stage during disinhibition-induced vocal repertoire expansion.

**Novel syllables are added at the end of stereotyped song motifs**

During baseline song recordings, ArchT+ birds produced $5.7 \pm 1.0$ (range: 4–7 syllables) different syllables per motif and after optogenetic stimulation, the birds increased the syllable number to $9.7 \pm 1.2$ (range: 8–11 syllables) (Fig. 4a and Supplementary Fig. 6). Therefore, we assessed whether the initial length of a birds' song measured as the

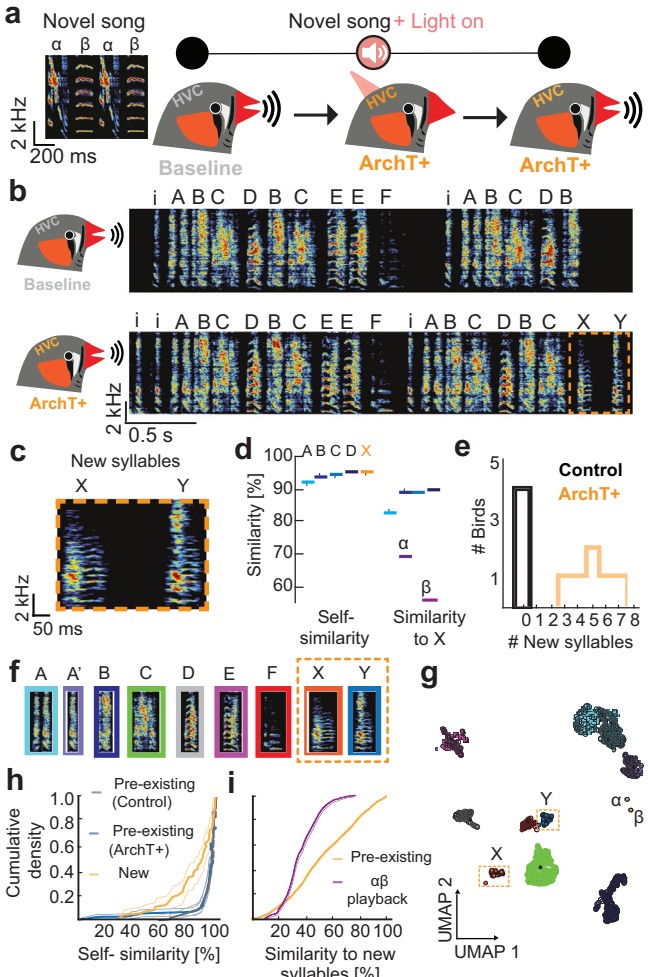

**Fig. 3 | Optogenetic silencing of HVC interneurons during playbacks leads to the production of novel song elements in adult zebra finches. a** Novel song playback composed of syllables α and β that is presented during optogenetic manipulation. Experimental timeline. The baseline song was recorded after the virus was expressed (6 weeks). Subsequently, birds received playbacks in conjunction with optogenetic silencing of transduced interneurons in HVC for 4 weeks. Afterwards, birds were kept in a soundbox without further manipulation and the final song was recorded after 4 weeks. **b** Song bout of an example ArchT+ bird during baseline (top panel) and at final stage (bottom panel). Individual syllables are marked with capital letters. 'i' indicates introductory notes. Novel syllables are highlighted with an orange dotted line. **c** Magnified spectrograms of the novel syllables X and Y. **d** Self-similarity score (left) and cross-similarity score with respect to the novel syllable ('X') (right). Source data are provided as a Source Data file. **e** Number of new syllables produced in control birds and in ArchT+ birds. **f** Example syllables from final song recordings. **g** UMAP plot with HDBscan clustering of all syllables sung during baseline (open circles) and during the final recording (filled circles). Novel syllables are highlighted with orange dotted boxes. **h** ECD curves of self-similarity comparisons for all syllables sung by control birds (grey) and ArchT+ birds (blue for syllables that were sung during baseline; $p = 0.7261$., yellow for novel syllables $p = 2.6548 \times 10^{-8}$. Overall difference: $\chi^2 = 34.4121$, $p = 3.37 \times 10^{-8}$, Kruskal–Wallis test. Bold lines highlight the means, thin lines indicate the 95% confidence interval for each data set (**i**) ECD distributions and 95% confidence intervals of cross-similarity of new syllables to baseline syllables (yellow) and to the α and β-playback syllables (purple). $\chi^2 = 780.0247$, $p = 1.19 \times 10^{-171}$, Kruskal–Wallis test. Bold lines highlight the means, thin lines indicate the 95% confidence interval for each data set.

number of different syllables within a motif influences their ability to acquire novel syllable elements. We correlated the number of different syllables produced by each bird during baseline recordings with the increase of different syllables after the experiments (Fig. 4a) and found

no significant correlation in controls (grey) or ArchT+ birds (yellow). However, the more syllables the initial song contained the fewer syllables were added which suggests that there might be a maximum capacity for a number of syllables a zebra finch can produce.

Next, we asked if the birds' song structure at the level of syllable sequence changed by measuring song linearity and consistency[40]. Song linearity did not differ across the three-time points (before manipulation (baseline), after four weeks with playback and optogenetic stimulations (light), after additional four weeks without playback or stimulations (final)) between ArchT+ and control birds (Fig. 4b). In contrast, song consistency was lower in ArchT+ birds after the light timepoint in relation to baseline and remained low during the final recording (Fig. 4c). These results that only consistency but not linearity is affected by silencing interneurons during playbacks of new songs implies that the pre-existing song sequence remained stable while occurrences of novel syllables were rare and restricted to a specific position within the song.

To explore the positioning of the novel syllables with respect to the learned song, we visualized individual syllable transition probabilities (Fig. 4d example and Supplementary Fig. 6 all birds). Birds in the control and ArchT+ groups, maintained their motifs' core sequence, as indicated by high transition probabilities between the pre-existing syllables in their original motif, irrespective of pre- or post-playback recordings (Fig. 4d). Novel syllables were most frequently added towards the end of the original motifs (Fig. 4d, e and Supplementary Figs. 7 and 8) after which a new rendition of the next song started. The distribution of the relative probability also remained stable across both recording epochs (Fig. 4d, e). These results suggest that the previously learned song sequence cannot be overwritten but instead, the addition of a new vocal motor skill can be accomplished.

## Discussion

In summary, by utilizing pharmacology (Fig. 1) and an optogenetic approach (Figs. 2 and 3) to manipulate inhibition in HVC, we reintroduced the capacity for vocal motor plasticity in adult zebra finches. This enabled birds to add new syllables to their existing "crystalized" songs (Fig. 4), at ages well beyond their critical period for vocal learning. Our viral construct selectively transfected inhibitory interneurons within a specific microcircuit[55] (Fig. 2) and allowed these neurons to be inactivated with temporally precise optogenetic stimulation, coinciding with particular auditory experiences. Following these manipulations, we found that adult zebra finches primarily added novel syllables (Fig. 3) to the ends of existing motifs (Fig. 4), demonstrating lasting expansion of their vocal repertoires, without degrading the performance of the previously consolidated songs.

Contrary to previous models[13] that predicted only the refinement of existing syllables upon hearing the tutor song again, we observed the emergence of new syllables in birds exposed to a novel two-syllable song. This unexpected acquisition of novel motor skills, diverging from the provided templates without altering existing vocalizations, could be attributed to several factors. The artificial playback song likely differed substantially from the birds' original tutor song, potentially triggering auditory filtering at early processing stages. This could limit HVC activity to responses matching acoustic features shared with the tutor song. Additionally, extensive vocal practice may impose physical constraints on the syringeal muscles[56], potentially hindering the production of novel syllables despite the altered neural activity. Finally, variations in viral effectiveness likely led to incomplete inhibition of HVC neurons, allowing some to remain active during stimulation, unpredictably gating auditory input and contributing to the observed variability. The finding that existing vocalizations remained unaffected after the optogenetic manipulation may be partially explained by the possibility that not all HVC premotor neurons are active during singing in adult zebra finches[57]. This raises the

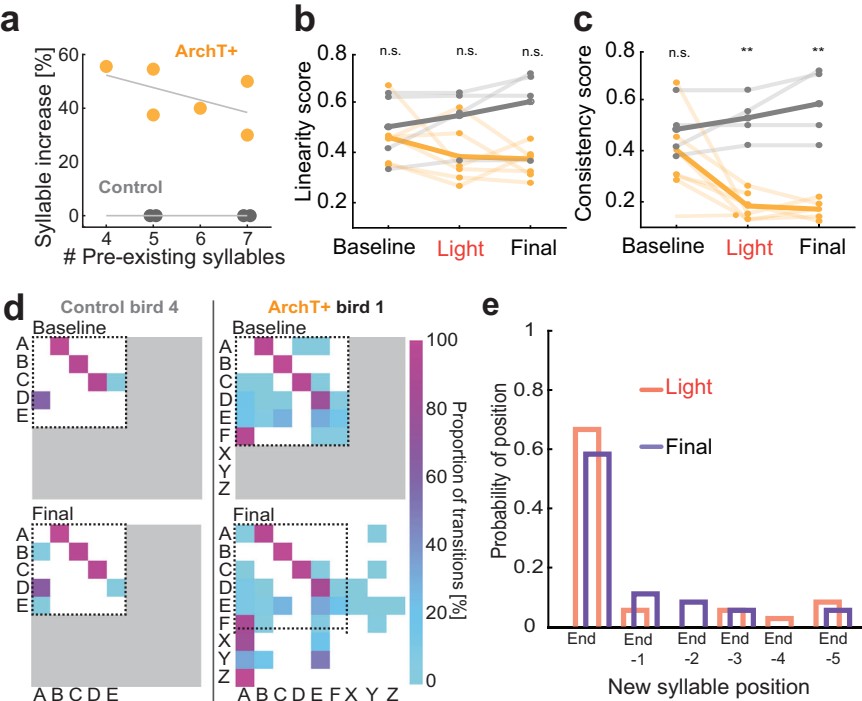

**Fig. 4 | Novel syllables are added at the end of stereotyped motifs. a** Percent syllable increase dependent on the number of pre-existing syllables (Control: $R^2 = 0$, $p = 1$; ArchT: $R^2 = 0.2951$, $p = 0.2653$, Pearson correlation). **b** Song linearity before manipulation (baseline), after four weeks of playbacks coupled to light stimulation (light) and after four additional weeks without further stimulation (final). Yellow shaded lines represent 6 individual ArchT+ birds, bold yellow line indicates the group mean. Grey lines represent 4 Control birds with their mean shown by the bold grey line. $p = 0.2592$, Friedman-test. **c** Song consistency (plotted for the same birds as in (**b**), $p = 0.006$, Friedman-test. **d** Syllable transition matrices for one control and one ArchT+ bird during baseline and final time points. Colour indicates proportion from low (turquoise) to high (purple). White shaded area indicates the range of possible transitions at the stage of recording based on the presence or absence of syllables. The dotted black line indicates the range covered by transitions during the baseline time point. Grey shaded area represents impossible transitions based on the occurrence of syllables. **e** Relative probability histogram for the positioning of new syllable occurrences across all ArchT+ birds in relation to the motifs' ends. Data are binned by sequence position for two test time points (after light stimulation (red); final stage (purple), $p = 0.8101$, Wilcoxon signed-rank test. Source data for (**a**–**c** and **e**) are provided as a Source Data file.

possibility that inactive neurons could potentially be recruited to create new sequences and generate novel syllables.

Critical periods are not limited to vocal learning but they also regulate the development of sensory systems. For instance, mice exhibit a temporary sensitivity to monocular deprivation, highlighting a critical period for visual plasticity in their ocular dominance development[16]. It has been demonstrated that GAD65 knockout mice are typically unresponsive to brief monocular deprivation since their visual system does not undergo a critical period. However, their critical phase for ocular dominance plasticity can be restored or extended by treatment with diazepam[21], which enhances inhibitory synaptic activity. This suggests that intracortical inhibition in these mice is perpetually at a level where plasticity could be triggered if given the right conditions. Similarly, it has been demonstrated that a critical period for social reward learning can be reopened through the administration of psychedelic drugs[22]. In this case, the window for enhanced social learning remains open throughout the duration of the treatment and for a varying period afterwards, depending on the specific drug used. Our pharmacological approach, mirroring the methodology and outcome observed in social systems, did not induce long-term plasticity. Instead, the observed effects appear to be solely attributable to acute modifications of the inhibitory network.

However, akin to observations in sensory systems, our optogenetic manipulation precisely targets inhibitory network activity exclusively during exogenous auditory input. This targeted approach achieves a more effective and long-lasting effect by allowing the neural network to learn the production of a new motor skill while ensuring that pre-existing motor production proceeds without disruption. If our optogenetic manipulations had extended beyond the period of auditory input into the time of song production, we would anticipate results similar to those observed with higher gabazine concentrations[39]. The song's spectral features might degrade, as seen in previous studies where HVC inhibition was perturbed with Gabazine, leading to either a degradation of song structure or a complete cessation of song production. Notably, no new syllables emerged in those studies, highlighting another crucial role for inhibition within the HVC in maintaining normal song production.

Some species of songbirds regain the ability to learn new syllables naturally, through seasonally-induced physiological changes. Canaries, for instance, are such 'open-ended learners' and can repeatedly learn new song syllables each mating period. This seasonal reopening of learning periods has been linked to systemic changes in testosterone level[58], influencing the structure of perineuronal nets that regulate the surrounding environment of inhibitory interneurons[16]. Our targeted manipulations of inhibition do not rely on such large systemic mechanisms and thus demonstrate that inhibitory processes on their own play a fundamental role in moderating the emergence of vocal motor programmes.

By highlighting the importance of local inhibition for adult motor skill learning in songbirds, this study raises new questions to be addressed by future research. For example, can the ability to reopen vocal motor critical periods through targeted manipulation of inhibition be applied more generally to retain or enhance learning abilities? Exploring such questions may have further implications for improving

recovery from neurological injuries, and advancing treatments for neurodevelopmental disorders.

## Methods

### Ethical statement
Animal housing and experimental procedures were performed in accordance with EU regulations and according to German national legislation on animal experimentation under licences AZ 02-19-153 and AZ 02-19-185 of the government of Upper Bavaria.

### Animal housing
All adult male zebra finches (>150 days post-hatch (dph)) were bred in the colony at the Max-Planck Institute for Biological Intelligence. Adult female zebra finches to accompany the experimental birds originated from the same breeding colony. All birds were reared in a breeding aviary ($200 \times 200 \times 200$ cm) and subsequently housed in single-sex aviaries ($100 \times 200 \times 200$ cm) at a 14:10 light:dark (LD) cycle. During the experiments, birds were kept in sound-attenuated boxes ($60 \times 60 \times 120$ cm).

### Animals for pharmacological manipulations
Birds were screened for reliable song production and only those individuals who immediately produced a song upon presenting a female bird were selected for experimentation. To ensure statistical power, only Birds that produced more than 20 motifs under PBS and gabazine conditions were considered for analysis. The total number of experimental birds analysed was 5 for the gabazine plus (+) playback group and 4 for the gabazine without (−) playback group. The age of birds that did not receive playbacks was $303 \pm 84$ dph (range: 159–365 dph) at the onset of pharmacological experiments. Birds that received playbacks during the infusion were $202 \pm 35$ dph (range: 155–207).

### Surgery for pharmacological manipulations
Birds were starved 30 min prior to the surgery and received an intra-pectoral injection of the analgesic Metamizol (100 mg/kg, Bela-pharm, Vechta). Then they were anaesthetized with isoflurane (1–3% in oxygen). The location of HVC was determined based on stereotactic coordinates (0.3 mm anterior, 2.3 mm lateral of the bifurcation of the midsagittal sinus) and confirmed with antidromic stimulation of the robust nucleus of the archiopallium RA[59]. A short biphasic current pulse of 100 μA was applied at 1 Hz for 5 ms in RA and was detected by a carbon fibre electrode (Carbostar-1, Kation Scientific, Minneapolis, MN) placed in the location of HVC. The locations at which the antidromic spikes could be detected were taken as the confirmed coordinates of HVC. Two bilateral, rectangular craniotomies sized $1 \times 0.5$ mm were made. Dura was removed after which cranial windows were covered with a silicone elastomer (Kwik-Cast; WPI, Sarasota, FL). Additionally, a custom-made stainless-steel head plate was implanted on the skull using dental cement (Paladur, Kulzer International, Hanau) for head-fixing the bird before each experimental session by applying either PBS or gabazine.

### Infusion of HVC with gabazine
Before the song recordings, the GABA-A receptor antagonist gabazine (SR-95531, Sigma Aldrich, St. Louis, MO) was applied bilaterally (0.005–0.01 mM in PBS (Carl Roth, Karlsruhe) onto HVC via saturated gel foam sponges (Avitene Ultrafoam, Bard, Murray Hill, NJ). The solution was warmed to 40 °C before application. For control experiments, PBS was used instead of gabazine. In order to restrict the area of application, wells of silicone elastomer were created around the craniotomies to keep the sponge in place. Before and after all experiments, craniotomies were cleaned of any overlying tissue and flushed with PBS warmed to 40 °C. The craniotomies were subsequently sealed with fresh silicone elastomer. Song produced by infused animals was recorded for 1 h after the application of either substance in the presence or absence of playbacks.

### Assessing the spread of a pharmacological agent into HVC
Fluorescence-conjugated Muscimol (M23400, Invitrogen, Waltham, MS) was applied onto HVC (0.01 mM in PBS) using the above-saturated sponge approach. After 1 h of application, the bird was perfused with ice-cold 4% paraformaldehyde (Sigma-Aldrich, St. Louis, MO) in 1× PBS. The brain was immersed in a 15% sucrose solution followed by a 30% sucrose solution (Sigma-Aldrich, St. Louis, MO) until saturation. The brain was then sliced (52 μm thickness) using a cryomicrotome. The slices were mounted and stained using Hoechst (H1399, Thermo Fisher Scientific, Waltham, MA). Images were generated using a Leica Fluorescence Microscope (Supplementary Fig. 2). While muscimol (114.10 g/mol) likely diffuses more widely than gabazine (287.31 g/mol), this approach provides an upper bound for the drug spread, confirming that areas surrounding HVC remain unaffected.

### Song recordings and stimulus preparation for pharmacological experiments
Songs were recorded in sound-attenuated chambers using custom MATLAB (v2018b, The Mathworks, Inc., Natick, MA) scripts at a sampling rate of 40 kHz. Songs were recorded before surgery to obtain playbacks for the experiments and for 1 h after the application of gabazine/PBS. A female zebra finch was presented briefly to motivate the male zebra finch to sing. Immediately after gel foam application and every 15 min thereafter, the birds were presented with a playback of the modified BOS for 2 min, for a total of four times in 1 h. Modified playbacks were generated from song motifs sung in the presence of a female zebra finch. Sequence changes were produced in Audacity 2.3.0 (audacityteam.org) by switching specific syllable elements within a bird's stereotyped motifs. As syllables can be identified by short gaps of silence (~50 ms), we interchanged the second and third syllables within a stereotyped motif. After the experiments, birds were sacrificed with an overdose of isofluran.

### Animals for optogenetic experiments
Prior to the start of optogenetics experiments, a focal male and one female companion bird were moved to cages ($120 \times 50 \times 50$ cm) in sound-attenuated chambers with a 14:10 LD cycle. Birds remained in these chambers for the entire duration of the experiment. Birds included in the optogenetics experiments as controls ($n = 4$) were $315 \pm 90$ dph (range: 209–423 dph) at the experiments' onset. Birds in the ArchT+ group ($n = 6$) were $226 \pm 35$ dph (range: 168–284 dph) when playback experiments began.

### Virus production
Plasmids for virus production were graciously provided by Prof. Dr. Peter Hegemann and modified at the Department of Behavioural Biology at the Freie Universität Berlin. pAAV-mDlx-ChR2-mCherry-Fishell-3 was a gift from Gordon Fishell (Addgene plasmid # 83898; http://n2t.net/addgene:83898; RRID: Addgene_83898)[42] to express ArchT[60] fused to TS-EYFP-ER sequence that was cloned from a pcDNA3.1 CMV-ChRmine-TS-EYFP-ER plasmid gift from the Hegemann laboratory at the Humboldt Universität. Control constructs consisted of the same backbone and reporter gene but did not contain ArchT. Plasmids of the constructs used in this study are available upon request.

### Surgery for virus injection and ferrule implantation
For viral injections and the implantation of the optic ferrules, birds were starved for 30 min prior to the surgery. Birds received an intrapectoral injection of the analgesic Metamizol (100 mg/kg, Bela-pharm, Vechta) and were anaesthetised with isoflurane (2%, DK Pharma, Bocholt) oxygen mixture (0.8 l/min) using a custom beak mask. Once

fully anaesthetised, birds were fitted into a stereotaxic apparatus (Kopf, Tujunga, CA). Feathers were removed at the surgical site and following disinfection (Betaisodona, Mundipharma, Frankfurt a. M.), the skin was numbed with topical Lidocaine cream (Aspen Pharma, Munich). After 5 min a 2 cm long incision was made along the midline of the scalp. Using a dental drill (Johnson-Promident, Valley Cottage, NY), the outermost layer of bone and the trabeculae were removed and the bifurcation of the midsagittal sinus ('lambda') was identified. Above HVC, which was located based on stereotactic coordinates relative to lambda, bilateral craniotomies (0.3 × 0.3 mm) were opened to allow for injections and implants.

For viral injections, 2.5 µl of viral particles (VP) with a titre of >6 × 10⁶ VP/ml were injected per hemisphere with a Nanoject V2 (Drummond Scientific Co, Broomall, PA, 2.3 nl steps, slow setting). The head angle was set to 70° and injections took place relative to the midsagittal sinus at 0.0 and 0.2 anterior/posterior and ±1.8, 2.0, and 2.2 medial/lateral at depths of 0.25 and 0.5 mm, respectively. Additional injections were performed at 0.35 mm depth at 0.1 mm anterior/posterior, ±1.6 and 2.4 mm medial/lateral, as well as at −0.2 mm and 0.4 mm anterior/posterior and ±2.0 mm medial/lateral. Following the injections, custom stainless-steel implant guides were fitted onto the craniotomies with a silicone polymer and fixated with dental cement. With the implant guides in place, custom ceramic optic fibre ferrules (0.39 NA, 200 um multimode fibre diameter, based on CFMXA05, Thorlabs, Newton, NJ) were inserted into the guides and fixated with a mixture of dental cement (Kulzer, Hanau) and superglue (Loctite, Hartford, CT). After the implant, the skin was repositioned with tissue glue (Vetbond, 3 M, Saint Paul, MS) around the implant sites and covered again with Lidocaine crème. Prior to the removal of the anaesthetic gas, the birds received an intrapectoral injection of meloxicam (0.2 mg/kg, Boehringer-Ingelheim, Ingelheim am Rhein). After surgery, birds were retransferred to their home cage. Experiments began after a recovery and viral incubation period of six weeks.

### Neural recordings and spike sorting
To validate if the ArchT construct prevents neurons from spiking under illumination, birds were injected with the viral construct as described above. No implant guides or ferrules were inserted, instead, the drill sites were filled with silicon elastomer (KauPo, Spaichingen) and skin was closed with tissue glue (Vetbond, 3 M, Saint Paul, MN) and the birds were moved back to their home cages for recovery.

Six weeks after injections, birds were anaesthetised as above and craniotomies were reopened. An optic fibre (0.39 NA, 200 µm MM, Thorlabs, Newton, NJ) was placed above HVC. A silicone probe (A1x16-Poly2-5mm-50s-177, Neuronexus, Ann Arbour, MI) coated in DiI crystals (ab145311, Abcam, Cambridge) was inserted into HVC. Signals were digitized at a rate of 30 kHz with a 16-channel RHD head stage amplifier acquired through a USB interface board (Intan Technologies, Los Angeles, CA) and saved onto a computer for offline analysis. For light stimulation during recordings, the LED driver (DC4100, Thorlabs, Newton, NJ) was controlled via an Arduino UNO (Arduino, Somerville, MA). Light intensity from Fibre-Coupled LEDs (M595F2ʰ, Thorlabs, Newton, NJ) was set to 2 mW/mm² at the recording site. Light stimuli were repeated 30 times in three blocks with a stimulus length of 100 ms an inter-stimulus interval of 200 ms and an inter-block interval of 2 s. Individual recorded units were identified based on waveform properties and principal component analyses (x64 V4, Offline Sorter, Plexon Inc., Dallas, TX).

### Song recordings prior to optogenetic experiments
After the incubation period of six weeks during the optogenetics experiments, a baseline song was recorded from each bird while the birds were tethered but not stimulated by custom optic fibres (based on FT200UMT, Thorlabs, Newton, NJ).

### Light stimulations in freely moving birds
During stimulation, birds were tethered to one 595 nm LED light source (M595F2ʰ, Thorlabs, Newton, NJ) per hemisphere which was driven by an LED driver (DC4100, Thorlabs, Newton, NJ). Every day birds received 200 playbacks paired with bilateral 595 nm light stimulation at an intensity of 2 mW/mm² during two separate blocks (9–11 AM; 5–7 PM). Each playback consisted of two motif repetitions, each containing two syllables (α and β, (Fig. 3a)[47]. Light stimulation lasted from 50 ms prior to sound onset until 50 ms after sound offset. After a playback period of four weeks, the song was recorded (light) and the birds remained in the experimental boxes for another four weeks without any playback stimulation to validate lasting song changes (final).

### Brain extractions after optogenetic experiments
After completion of the playback and post-playback phases (28 + 28 days), all birds received an overdose of isoflurane. Birds were immediately perfused with ice-cold paraformaldehyde (4% PFA, Sigma-Aldrich, St. Louis, MO) in 1× PBS (Carl Roth, Karlsruhe). Brains were extracted and post-fixated for 24 hrs in 4% PFA in 1× PBS at 4 °C and afterwards transferred into 1× PBS.

### Immunohistochemistry
Fixated brains were sliced to 50 µm on a vibratome (VT1200S, Leica, Wetzlar). For validation of specific viral expression and targeted injection, slices were incubated with primary antibodies against PV (1:1000, mouse, PV 246, Swant, Burgdorf) and GFP (1:1000, rabbit, ab290, Abcam, Cambridge). Secondary incubations were conducted with donkey anti-mouse (1:200, A1037, Invitrogen, Waltham, MS), and donkey anti-rabbit (1:200, A1042, Invitrogen) antibodies, respectively.

For the visualisation of a broader population of inhibitory interneurons within HVC, slices were incubated with primary antibodies against PV, CR (1:1000, goat, CG1, Swant, Burgdorf), CB (1:1000, rabbit, CB38, Swant, Burgdorf), and eYFP (1:400, chicken, A10262, Thermo Fisher Scientific, Waltham, MS) and the matching secondaries for eYFP (goat anti-chicken, 1:200, 103545155, Dianova, Hamburg), PV (same as above), CR (donkey anti-goat, 1:200, A11058, Invitrogen), and CB (donkey anti-rabbit, 1:200, A1042, Invitrogen). Nuclear counterstains were done with Hoechst (H1399, Thermo Fisher Scientific). Following staining, slices were mounted and photographed with a confocal microscope (SP8, Leica, Wetzlar).

### Assessment of labelled cells
To analyse the proportion of targeted interneurons infected in HVC, the number of eYFP-expressing neurons was quantified among PV-positive cells. For a more complete analysis of the targeted inhibitory interneuron population the amount of either PV, CR, or CB positive neurons that also expressed the fluorophore eYFP were quantified for a subset of samples. This approach has been shown to cover nearly the entire population of inhibitory interneurons[46].

### Data analyses and statistics
Songs were labelled and analysed using custom scripts in MATLAB (v2020a, The Mathworks, Inc., Natick, MA). The syllables of the first 50 song bouts at the recording epoch were segmented using automated thresholds and were labelled manually after third-person blinding of data. Changes in sequence were detected and analysed using a custom MATLAB script, which calculates the probabilities of all possible syllable transitions. Song similarity scores and spectral parameters of recordings were calculated with SAP2011[48]. Similarity score comparisons were conducted symmetrically across the time course of each pair of syllables or motifs. A syllable was considered to be significantly different if the similarity score diverged 1.96 × standard deviation from the similarity score calculated from pre-existing syllables.

For the detection of syllable sequence changes in the birds' songs, previously established measures were used[40]. Sequence Linearity measures the degree of branching of the song i.e. whether the bird's song linearly follows a stereotypical sequence.

$$Sequence\ linearity = \frac{\#different\ notes\ per\ song}{\#transition\ types\ per\ song} \quad (1)$$

For a highly linear song, sequence linearity = 1.

Sequence consistency measures the number of times the song follows the stereotypical original path.

$$Sequence\ Consistency = \frac{\Sigma\ typical\ transitions\ per\ song}{\Sigma\ total\ transitions\ per\ song} \quad (2)$$

For a highly consistent song, sequence consistency = 1.

Statistical analyses were conducted with non-parametric tests. Statistical analyses and tests were conducted in MATLAB and reported individually. All data are reported in the form of means ± standard deviations unless noted otherwise. Significance levels are indicated by asterisks within each panel and defined by the size of the $p$-value ($*p < 0.05$, $**p < 0.01$, $***p < 0.001$).

To confirm the existence of novel syllables, unbiased by choice of specific acoustic features, we employed a supervised deep learning algorithm[49] to generate a model which recognises syllables produced by all zebra finches that participated in the optogenetics experiments ($N_{Control} = 4$, $N_{ArchT+} = 6$). We used ten exemplary motifs for each bird as segmented training data. Once the model reached high performance (precision: 0.9035, recall: 0.8055), we segmented all syllables that each bird produced during one day of either baseline or after light stimulation or final recording (basic settings, min. syllable length = 5 ms). Based on this syllable segmentation, UMAPs were created with custom Jupyter notebooks to visualise the distribution of the segmented syllables across a high-dimensional space. HDBscan was applied for clustering the data[51].

## Reporting summary

Further information on research design is available in the Nature Portfolio Reporting Summary linked to this article.

## Data availability

Source data are provided with this paper. Example data are available here https://github.com/vallentinlab/HVC_disinhibtion. Due to space limitations of the public repository, the complete song and neural recordings will be made available upon request. Please contact the corresponding author. Source data are provided with this paper.

## Code availability

All custom scripts including example data are available here https://github.com/vallentinlab/HVC_disinhibtion.

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

## Acknowledgements

We thank Constance Scharff for insightful comments on the manuscript and for her generous scientific and life support. We also thank Tamir Eliav, Jonathan Benichov and current members of the Vallentin Lab for helpful feedback on previous versions of the manuscript. We are thankful to Barbara Buhlmann and all caretakers for excellent animal care. We would also like to thank Andrea Yinth Bernal Sierra, Johannes Vierock and Peter Hegemann for the generous gifting of plasmids, Jannis Hildebrandt for helpful advice on the optogenetics setup, Jan Clemens for trouble-shooting DAS and Ulla Kobalz for tedious cloning experiments. Additionally, we would like to acknowledge the support of the Viral Core Facility (VCF-Charite) and the Core Facility BioSupraMol, which is funded by the DFG. This work was supported by the HORIZON EUROPE European Research Council (ERC)-2017-StG-757459 MIDNIGHT, the Deutsche Forschungsgemeinschaft VA742/2-1 and the Deutsche Forschungsgemeinschaft 327654276–SFB 1315—awarded to D.V.

## Author contributions

F.H. and D.V. conceived the study and designed the experiments, F.H., E.M., A.K., and D.V. conducted the experiments, E.M. and F.H. designed the viral constructs, E.M. and A.K. performed immunohistochemical procedures and microscopy, F.H., E.M., A.K., and D.V. analysed the data, F.H. and D.V. wrote the first draft of the manuscript, all authors participated in writing and editing of the manuscript, D.V. acquired funding and supervised the project.

## Funding

## Competing interests

The authors declare no competing interests.
