## [Peer Review File · Nature Communications]

Disinhibition enables vocal repertoire expansion after a critical periodREVIEWER COMMENTS

Reviewer #1 (Remarks to the Author):

In this important manuscript, Heim et al. elegantly demonstrate how disinhibiting the premotor region in the zebra finch brain (HVC) while simultaneously playing back a novel song drove vocal plasticity in birds well past the critical period. Specifically, they find that disinhibiting via optogenetics led to the concatenation of new syllables at the end of the original song, with no change in the expert song. This intriguing result suggests that other mechanisms maintain the original song fidelity – there was no evidence of any disruption to the existing song – while inhibitory processes likely play a role in limiting the length or complexity of a given song. The figures and writing are very clear and appropriate caveats are made when required. These data provide a major advance in our understanding of vocal motor plasticity in the zebra finch with implications for other species and warrants publication in Nature Communications, with only minor revisions to improve the readability of the manuscript and to better contextualize the manuscript:

1) Previous work from Vallentin et al. demonstrated the critical importance of inhibition in the HVC during the development of the song. This work makes some predictions about the role of inhibition in adult animals from Figure S10C where inhibitory network changes in HVC are correlated with song performance rather than age. In one framing, this work provides a possible explanation for why the original song does not change but does not predict the finding that additional syllables are added. I believe that somehow incorporating these ideas into the intro or discussion would help strengthen this manuscript and put it into the context of previous work.

2) An important implication of this work is the relationship between critical periods in other systems (sensory/social) and motor critical periods. It might be helpful to more explicitly compare/contrast the current work with studies from re-opening the critical period in adult animals in sensory studies (Takao Hensch et al.) and social behavior (Nardou et al.) This could, for example, go into the discussion.

3) The authors justify (correctly) that optogenetic (versus pharmacological) disinhibition provides an opportunity to isolate whether these effects are due to playback exposure or song production. They demonstrate that the effects described (addition of new syllables) is specific to playback exposure. This is a solid and empirically grounded finding. It could be useful to speculate (no new experiments or analysis) in a sentence or two in the discussion what might happen if optogenetic inhibition of inhibitory neurons in HVC (circuit disinhibition) at the time of song production might do. Just to clarify, I do not believe any additional experiments are needed as that will be a full study itself.

Reviewer #2 (Remarks to the Author):

In the present study, Heim and colleagues investigated a long-standing neurobiological question, namely, whether the critical temporal phases crucial for vocal learning in songbirds can be reactivated after their closure. Although it has been shown that inhibitory mechanisms are involved in the closure of sensitive phases for song learning, it has never been demonstrated whether such critical phases for song learning can be reopened by manipulating these inhibitory neurons. Using a number of sophisticated, state-of-the-art techniques in a highly systematic way, the authors showed for the first time that lifting inhibition in HVC only during periods of presentation of new song syllables allows adult birds to produce new, syllables, not previously produced, thereby, expanding their vocal repertoire during this reintroduced phase of vocal-motor plasticity. These new syllables were added to existing song motifs without altering the original song sequence, demonstrating a selective effect of inhibition on vocal plasticity. The present study highlights the contribution of inhibitory factors to the closure of the vocal motor critical period, which limits the ability of adult birds to learn new syllables and suggests that local inhibitory processes play a fundamental role in moderating the emergence of vocal motor programs.

Overall, this is an impressive and highly original study! The methodological approach is well executed, and the data are convincing, well analyzed, and concisely presented. The manuscript is well-written and a pleasure to read! The findings raise questions about the potential applications of targeted manipulation of inhibition in the maintenance or enhancement of learning abilities, which might have implications for neurological recovery and the treatment of neurodevelopmental disorders. The paper is therefore of general interest and will clearly appeal to the large and diverse readership of Nature Communications.

I think this manuscript is almost ready to go. I just have a few comments that the authors might want to address to improve their manuscript:

1. Why do some birds produce more than two new syllables in response to the acoustic presentation of the two templates? Furthermore, why did not any bird produce a perfect match to one of the presented target syllables? Might this be due to a potentially incomplete bilateral activation of HVC inhibition?
2. Figure 1g: Please add explanations for the black and green curves.
3. Extended Data Figure 3c should be included in one of the main figures as it is an important part of the experimental approach.

Reviewer #3 (Remarks to the Author):

The study by Heim et al. elegantly investigates the possibility that inhibiting inhibitory neurons in HVC reopens the window of plasticity for song learning in adult male zebra finches. First, the authors show that pharmacological suppression of GABA-A receptors with gabazine, leads to adding new syllables to the adult song. Then, the authors use optogenetics to suppress inhibitory neurons in HVC while new songs are being played, for 4 weeks (200 daily playbacks). After this, the authors find that birds add novel syllables to their song, that are more similar to their own syllables than with the artificial ones, and are more likely to occur at the end of the song motif rather than interrupting it. The findings from this paper are relevant for song learning in birds, language learning in adult humans, and more broadly to neuroplasticity and recovery fields.

The authors do a fantastic job writing the paper, presenting and analyzing the data, and interpreting their results.

Minor observations:

- Do the new syllables added during gabazine infusion persist after the drug is washed out?
- In Figure 2, did the authors identify neurons that are actually disinhibited (activated) during suppression of inhibitory neurons. Given the title of the manuscript, it would be good to show that there is in fact disinhibition in HVC
- In the pharmacology experiment, the song is recorded while GABA-A antagonist is present, looking at acute effects. In the optogenetic experiment the recordings are done long after the inhibitory neurons are manipulated, looking at plasticity. Could the authors discuss these design differences and interpretations in their manuscript?
- In EDF 2, it is great that the authors look at drug diffusion but would the diffusion of gabazine and muscimol be equivalent? A very minor observation, that can be addressed in their Methods section.

Reviewer #4 (Remarks to the Author):

In this manuscript, Heim et al. induce vocal plasticity in a songbird beyond the critical period by manipulating inhibitory activity in the premotor area HVC. In a Nature paper, the manuscript's senior author has demonstrated that inhibition in HVC is stronger for learned syllables. This manuscript now employs clever pharmacological and optogenetic experiments to remove inhibition in HVC during the playback of novel song syllables. The authors demonstrate that disinhibition of HVC in adult birds alters song. Notably, after without inhibition, even brief exposure to novel song induces vocal motor plasticity, suggesting that the system retains the potential for change in adults. The authors further reveal that this plasticity is constrained: syllables from the

existing song change minimally, while novel syllables tend to be appended to the end of the song sequence, rather than inserted in the beginning or middle. This suggests that factors beyond inhibition "protect" learned syllables, and that inhibition largely prevents the acquisition of novel song elements.

The authors' conclusions are well supported by the data, and the results are highly original and relevant beyond the field of songbird research. The manuscript is a pleasure to read; the figures are clear, and the experimental and analysis methods are sound. I have a few comments—mainly regarding a more extensive discussion of the findings—that I am convinced Heim et al. will be able to address.

Major comments

Interpretation of the effects in the pharmacological experiments:

The authors present results from two types of experiments: short-term removal of inhibition using pharmacology, and long-term removal of inhibition using optogenetics. The two experiments produce slightly different results, suggesting that they reflect the effect of different processes.

The optogenetics experiments involve disinhibition and exposure to novel song for multiple weeks, during which more permanent structural changes in the vocal motor pathway are likely to occur. By contrast, in the pharmacological experiments, exposure to playback is very short: if I understand correctly, 4x2 minutes of playback within 1 hour is sufficient to induce plasticity during the pharmacological experiments. Additionally, in some birds, song changes even in the absence of playback.

Given the short timescale and the sparse playback, can one even talk of plasticity (changes in the circuit) here? Or does disinhibition simply release latent motor outputs that are normally suppressed by inhibition? The authors address this by comparing changes in song with and without playback. And the fact that changes in song persist even after the week-long optogenetic stimulation also demonstrates that disinhibition induces long-term changes in the circuit. But can the authors perform additional analyses to further discriminate acute from long-term effects of disinhibition? For instance, by tracking changes in song during the week-long optogenetic stimulation. Do the novel syllables change at all during the 4 weeks of stimulation? Or do they pop up at the beginning and are then simply appended to the existing song, without changes in their spectrotemporal properties? These analyses would shed light on the types of plasticity that the disinhibition can induce in the adult. If these analyses are not feasible, for instance because song was not recorded during the optogenetic stimulation phase, because the new song elements are too variable, or because of other limitations, then these issues should at least be addressed in the discussion.

Acute effects of disinhibition during singing:

It appears that disinhibition has surprisingly little direct effect on the existing song. One would expect a more substantial degradation in vocal performance when removing inhibition from HVC. Does this small effect simply reflect the limited scope of their manipulation? Or does it imply that inhibition in HVC is dispensable during vocal production? This should be discussed.

Relation between playback and changes in song:

The content of the playback does not seem to relate to how the song changes. In the pharmacological experiments, the order of two syllables is changed in the playback; during optogenetics, two new syllables are played back. Each time, the resulting changes in song have little relation to the content of the playback. It seems as if playback is simply required to induce plasticity, but playback does not shape the changes in vocal motor output. Moreover, in the pharmacological experiments, playback is not strictly necessary to change the song.

Can the authors discuss the role of playback in their experiments? How does the playback induce changes in song? Why does the content of the playback have so little impact on the song changes? In other words, what prevents adult birds from learning the new song? The authors briefly address this in line 296, and I am aware that it is impossible to answer these questions at this stage, but I believe the issue deserves a more detailed discussion.

Minor:

Typo in line 163: "presence or absence"

Point-by-point responses to the comments of the four reviewers (*reviewers' comments in black italics, our responses in green*) to the manuscript 'Disinhibition enables vocal repertoire expansion after a critical period' by Fabian Heim, Ezequiel Mendoza, Avani Koparkar and Daniela Vallentin.

Reviewer #1 (Remarks to the Author):

In this important manuscript, Heim et al. elegantly demonstrate how disinhibiting the premotor region in the zebra finch brain (HVC) while simultaneously playing back a novel song drove vocal plasticity in birds well past the critical period. Specifically, they find that disinhibiting via optogenetics led to the concatenation of new syllables at the end of the original song, with no change in the expert song. This intriguing result suggests that other mechanisms maintain the original song fidelity – there was no evidence of any disruption to the existing song – while inhibitory processes likely play a role in limiting the length or complexity of a given song. The figures and writing are very clear and appropriate caveats are made when required. These data provide a major advance in our understanding of vocal motor plasticity in the zebra finch with implications for other species and warrants publication in Nature Communications, with only minor revisions to improve the readability of the manuscript and to better contextualize the manuscript:

We greatly appreciate the positive feedback provided by Reviewer #1 for our manuscript. In the following response, we aim to address the suggested remarks and clarify certain aspects to enhance the readability and contextualization of our manuscript.

1) Previous work from Vallentin et al. demonstrated the critical importance of inhibition in the HVC during the development of the song. This work makes some predictions about the role of inhibition in adult animals from Figure S10C where inhibitory network changes in HVC are correlated with song performance rather than age. In one framing, this work provides a possible explanation for why the original song does not change but does not predict the finding that additional syllables are added. I believe that somehow incorporating these ideas into the intro or discussion would help strengthen this manuscript and put it into the context of previous work.

We agree with Reviewer #1 that a clearer comparison to previous work is necessary. We have revised the introduction to explicitly state the predictions of the earlier model (line 72-81) and discuss our experimental findings in that context, highlighting the discrepancies between the predicted and observed outcomes (line 459-474).

2) An important implication of this work is the relationship between critical periods in other systems (sensory/social) and motor critical periods. It might be helpful to more explicitly compare/contrast the current work with studies from re-opening the critical period in adult animals in sensory studies (Takao Hensch et al.) and social behavior (Nardou et al.) This could, for example, go into the discussion.

We appreciate Reviewer #1's insightful comment. While we cited seminal studies on the closure and reopening of sensory/social critical periods in the introduction, we neglected to draw explicit comparisons in the discussion. We have now revised the discussion to directly compare our findings with those from other modalities and animal models (line 476-491).

3) The authors justify (correctly) that optogenetic (versus pharmacological) disinhibition provides an opportunity to isolate whether these effects are due to playback exposure or song production. They demonstrate that the effects described (addition of new syllables) is specific to playback exposure. This is a solid and empirically grounded finding. It could be useful to speculate (no new experiments or analysis) in a sentence or two in the discussion what might happen if optogenetic inhibition of inhibitory neurons in HVC (circuit disinhibition)

at the time of song production might do. Just to clarify, I do not believe any additional experiments are needed as that will be a full study itself.

Reviewer #1's suggestion to speculate on the potential effects of optogenetic manipulation of inhibition during adult birdsong was intriguing. We have expanded our discussion to include insights from a previous study in which HVC inhibition was pharmacologically perturbed with a higher Gabazine concentration during singing. We hypothesize that similar impairments would likely emerge if optogenetic stimulation were extended to this period. This addition (lines 497-503) provides valuable context for the interpretation of our findings.

Reviewer #2 (Remarks to the Author):

In the present study, Heim and colleagues investigated a long-standing neurobiological question, namely, whether the critical temporal phases crucial for vocal learning in songbirds can be reactivated after their closure. Although it has been shown that inhibitory mechanisms are involved in the closure of sensitive phases for song learning, it has never been demonstrated whether such critical phases for song learning can be reopened by manipulating these inhibitory neurons. Using a number of sophisticated, state-of-the-art techniques in a highly systematic way, the authors showed for the first time that lifting inhibition in HVC only during periods of presentation of new song syllables allows adult birds to produce new, syllables, not previously produced, thereby, expanding their vocal repertoire during this reintroduced phase of vocal-motor plasticity. These new syllables were added to existing song motifs without altering the original song sequence, demonstrating a selective effect of inhibition on vocal plasticity. The present study highlights the contribution of inhibitory factors to the closure of the vocal motor critical period, which limits the ability of adult birds to learn new syllables and suggests that local inhibitory processes play a fundamental role in moderating the emergence of vocal motor programs.

Overall, this is an impressive and highly original study! The methodological approach is well executed, and the data are convincing, well analyzed, and concisely presented. The manuscript is well-written and a pleasure to read! The findings raise questions about the potential applications of targeted manipulation of inhibition in the maintenance or enhancement of learning abilities, which might have implications for neurological recovery and the treatment of neurodevelopmental disorders. The paper is therefore of general interest and will clearly appeal to the large and diverse readership of Nature Communications.

I think this manuscript is almost ready to go. I just have a few comments that the authors might want to address to improve their manuscript:

We are deeply grateful to Reviewer #2 for their positive feedback regarding our study. We are confident that we can address the remaining comments in a manner that meets their expectations.

1. *Why do some birds produce more than two new syllables in response to the acoustic presentation of the two templates? Furthermore, why did not any bird produce a perfect match to one of the presented target syllables? Might this be due to a potentially incomplete bilateral activation of HVC inhibition?*

We agree with Reviewer #2 that the raised questions are important and warrant further investigation. While we do not yet have definitive answers, we have added a speculative discussion point that may stimulate future research in this area (line 459-474).

2. *Figure 1g: Please add explanations for the black and green curves.*

We apologize for the oversight in the figure legend. We have revised it to ensure all lines are accurately labelled and reflect their respective representations (Figure 1g, Figure legend, line 144-149).

3. *Extended Data Figure 3c should be included in one of the main figures as it is an important part of the experimental approach.*

We appreciate Reviewer #2's suggestion and have incorporated the song playback presented during optogenetic stimulation into the main Figure 3A.

Reviewer #3 (Remarks to the Author):

The study by Heim et al. elegantly investigates the possibility that inhibiting inhibitory neurons in HVC reopens the window of plasticity for song learning in adult male zebra finches. First, the authors show that pharmacological suppression of GABA-A receptors with gabazine, leads to adding new syllables to the adult song. Then, the authors use optogenetics to suppress inhibitory neurons in HVC while new songs are being played, for 4 weeks (200 daily playbacks). After this, the authors find that birds add novel syllables to their song, that are more similar to their own syllables than with the artificial ones, and are more likely to occur at the end of the song motif rather than interrupting it. The findings from this paper are relevant for song learning in birds, language learning in adult humans, and more broadly to neuroplasticity and recovery fields.

The authors do a fantastic job writing the paper, presenting and analyzing the data, and interpreting their results.

We sincerely thank Reviewer #3 for their encouraging feedback on our study. We are committed to addressing the remaining comments to their satisfaction.

Minor observations:

- Do the new syllables added during gabazine infusion persist after the drug is washed out?

We thank Reviewer #3 for raising this interesting question. All observed song changes induced by gabazine application were transient and returned to baseline levels following the drug washout period. This information is now included in the main text (line 123-125). We also now explicitly compare this finding to the result from the optogenetic experiment (line 488-497).

- In Figure 2, did the authors identify neurons that are actually disinhibited (activated) during suppression of inhibitory neurons. Given the title of the manuscript, it would be good to show that there is in fact disinhibition in HVC.

We appreciate Reviewer #3 raising this point. When recording from putative ArchT-expressing HVC neurons during light stimulation, we observed some neurons that were disinhibited. This observation has been clarified in the main text (lines 209-212) and is further illustrated in Supplementary Figure 3.

- In the pharmacology experiment, the song is recorded while GABA-A antagonist is present, looking at acute effects. In the optogenetic experiment the recordings are done long after the inhibitory neurons are manipulated, looking at plasticity. Could the authors discuss these design differences and interpretations in their manuscript?

We acknowledge Reviewer #3's insightful point. Recognizing the limitations of our initial pharmacological experiment, specifically the inability to precisely control the timing of inhibition, we developed the viral approach to address the impact of auditory input on the HVC circuitry. We have now explicitly outlined these differences in the discussion section (lines 476-497), where we also compare our results to other studies investigating the reopening of critical periods across various modalities and animal models.

- In EDF 2, it is great that the authors look at drug diffusion but would the diffusion of gabazine and muscimol be equivalent? A very minor observation, that can be addressed in their Methods section.

We agree with Reviewer #3 that tissue diffusion is variable across different drugs. To our knowledge fluorescently conjugated gabazine is unavailable. We therefore estimated the spread using muscimol (114.10 g/mol), which has a lower molecular weight than gabazine (287.31 g/mol). Given that muscimol is likely to diffuse more widely than gabazine, we can confidently conclude that gabazine application remains localized within the premotor nucleus HVC. This has been clarified in the Methods section (lines 579-581).

Reviewer #4 (Remarks to the Author):

In this manuscript, Heim et al. induce vocal plasticity in a songbird beyond the critical period by manipulating inhibitory activity in the premotor area HVC. In a Nature paper, the manuscript's senior author has demonstrated that inhibition in HVC is stronger for learned syllables. This manuscript now employs clever pharmacological and optogenetic experiments to remove inhibition in HVC during the playback of novel song syllables. The authors demonstrate that disinhibition of HVC in adult birds alters song. Notably, after without inhibition, even brief exposure to novel song induces vocal motor plasticity, suggesting that the system retains the potential for change in adults. The authors further reveal that this plasticity is constrained: syllables from the existing song change minimally, while novel syllables tend to be appended to the end of the song sequence, rather than inserted in the beginning or middle. This suggests that factors beyond inhibition "protect" learned syllables, and that inhibition largely prevents the acquisition of novel song elements.

The authors' conclusions are well supported by the data, and the results are highly original and relevant beyond the field of songbird research. The manuscript is a pleasure to read; the figures are clear, and the experimental and analysis methods are sound. I have a few comments—mainly regarding a more extensive discussion of the findings—that I am convinced Heim et al. will be able to address.

We are deeply grateful to Reviewer #4 for their positive feedback regarding our study. We believe that by addressing the remaining comments, we can further strengthen and improve our study, making it even more valuable for the scientific community.

Major comments

Interpretation of the effects in the pharmacological experiments:

The authors present results from two types of experiments: short-term removal of inhibition using pharmacology, and long-term removal of inhibition using optogenetics. The two experiments produce slightly different results, suggesting that they reflect the effect of different processes.

The optogenetics experiments involve disinhibition and exposure to novel song for multiple weeks, during which more permanent structural changes in the vocal motor pathway are likely to occur. By contrast, in the pharmacological experiments, exposure to playback is very short: if I understand correctly, 4x2 minutes of playback within 1 hour is sufficient to induce plasticity during the pharmacological experiments. Additionally, in some birds, song changes even in the absence of playback.

We absolutely agree with Reviewer #4 regarding the differing outcomes of the pharmacology and optogenetic experiments. This distinction is now more explicitly emphasized in the discussion (lines 488-497), highlighting the contrasting timescales and temporal precision of the perturbations' respective effectiveness. We elaborate further on these differences below.

Given the short timescale and the sparse playback, can one even talk of plasticity (changes in the circuit) here? Or does disinhibition simply release latent motor outputs that are normally suppressed by inhibition?

We thank Reviewer#4 for these valuable questions which we attempt to answer in the following: While our pharmacological approach did not result in long-term plasticity, the observed short-term changes in song production suggest that some modifications may persist, albeit below the threshold for causing a behavioral change. We agree with Reviewer #4 that disinhibition could potentially unveil differences in motor output typically masked by

general suppression. We have now incorporated this possibility into the manuscript (lines 123-125).

The authors address this by comparing changes in song with and without playback. And the fact that changes in song persist even after the week-long optogenetic stimulation also demonstrates that disinhibition induces long-term changes in the circuit. But can the authors perform additional analyses to further discriminate acute from long-term effects of disinhibition? For instance, by tracking changes in song during the week-long optogenetic stimulation. Do the novel syllables change at all during the 4 weeks of stimulation? Or do they pop up at the beginning and are then simply appended to the existing song, without changes in their spectrotemporal properties? These analyses would shed light on the types of plasticity that the disinhibition can induce in the adult. If these analyses are not feasible, for instance because song was not recorded during the optogenetic stimulation phase, because the new song elements are too variable, or because of other limitations, then these issues should at least be addressed in the discussion.

We agree with Reviewer #4 on the importance of addressing the differences in observed plasticity between the pharmacological and optogenetic experiments. Unfortunately, due to limited monitoring of only a subset of birds during optogenetic manipulation, we cannot draw robust conclusions from the available data. Thus, we can only report that new syllables emerged after the manipulation and were not always appended, as illustrated in Figure 4d,e and Supplementary Figure 7. However, the integration of these new syllables persisted throughout the four-week observation period.

We have now included a discussion of these observed differences in the manuscript (lines 488-497), placing them in the broader context of other studies investigating critical period reopening across various animal models and modalities.

Acute effects of disinhibition during singing:

It appears that disinhibition has surprisingly little direct effect on the existing song. One would expect a more substantial degradation in vocal performance when removing inhibition from HVC. Does this small effect simply reflect the limited scope of their manipulation? Or does it imply that inhibition in HVC is dispensable during vocal production? This should be discussed.

We acknowledge Reviewer #4's point regarding the potential impact of even low gabazine concentrations on song production. While a previous study (Kosche et al., Figure 2) showed that higher concentrations disrupt song structure, the absence of an effect at the lower concentration used here cannot be completely discounted. It is possible that subtle changes induced by gabazine, while not overtly affecting song production, could still influence the results of our pharmacological experiment. To address this limitation, we developed the viral approach, which allowed us to disentangle motor production from changes induced by auditory input during periods of reduced inhibition. This approach provides a clearer picture of the effects of disinhibition on vocal plasticity, as it eliminates the confounding factor of gabazine's direct influence on song production.

We have now explicitly incorporated this discussion into the manuscript (lines 488-503), highlighting the limitations of the pharmacological approach and the advantages of the viral approach in addressing this issue.

Relation between playback and changes in song:

The content of the playback does not seem to relate to how the song changes. In the pharmacological experiments, the order of two syllables is changed in the playback; during

optogenetics, two new syllables are played back. Each time, the resulting changes in song have little relation to the content of the playback. It seems as if playback is simply required to induce plasticity, but playback does not shape the changes in vocal motor output. Moreover, in the pharmacological experiments, playback is not strictly necessary to change the song.

Can the authors discuss the role of playback in their experiments? How does the playback induce changes in song? Why does the content of the playback have so little impact on the song changes? In other words, what prevents adult birds from learning the new song? The authors briefly address this in line 296, and I am aware that it is impossible to answer these questions at this stage, but I believe the issue deserves a more detailed discussion.

We appreciate Reviewer #4's feedback and apologize for not having addressed this concern earlier. We have now expanded the discussion section (lines 459-474) to elaborate on potential reasons why the birds did not directly imitate the playback.

We propose the following explanations for our results:

Auditory Filtering: The artificial playback song likely differed from the birds' original tutor song, potentially causing auditory filtering at early processing stages. This could limit HVC activity to responses to acoustic features shared with the tutor song. While testing this hypothesis directly would require the original tutor songs, it aligns with the observed results.

Physical Constraints: Extensive vocal practice might impose physical limitations on the syringeal muscles (Adam et al., 2024). Such constraints could hinder the production of novel syllables, even if neural activity is altered.

Incomplete Inhibition: Variations in viral effectiveness likely led to incomplete inhibition of HVC neurons. This could allow some neurons to remain active during stimulation, unpredictably gating auditory input and contributing to the observed variability.

Minor:

Typo in line 163: "presence or absence"

Changed.

REVIEWERS' COMMENTS

Reviewer #1 (Remarks to the Author):

The authors have done an excellent job responding to the minor suggestions I provided. I congratulate the authors on an excellent manuscript that provides a major conceptual advance in the field and look forward to seeing this in print!

Reviewer #2 (Remarks to the Author):

The authors have done an excellent job revising their manuscript. I therefore recommend that the manuscript be accepted for publication in Nature Communications. This is a great piece of work!

Reviewer #3 (Remarks to the Author):

The authors have addressed all my concerns and I have no further comments. From my point of view, this manuscript is ready for publication.

Reviewer #4 (Remarks to the Author):

The authors have done a fantastic job addressing all of the reviewers' comments. I recommend accepting this manuscript for publication.

A point-by-point response to the reviewers' comments:

We thank all four reviewers for their positive assessment of our work. We appreciate your careful consideration and constructive feedback, which have significantly improved the quality of our work.